# MOLECULAR ACTIVE LEARNING: CAN LLMS HELP?

## ABSTRACT

Drug discovery, and molecular discovery more broadly, can be framed as a sequential active learning problem —facing a candidate pool, strategies are designed to sequentially acquire molecules to assay, aiming to find the best molecule within the fewest rounds of trial and error. To automate this process, Bayesian optimization (BO) methods can mimic the approach of human medicinal chemists by constructing *representations* from existing knowledge, quantifying *uncertainty* for the predictions, and designing *acquisition* experiments that balance exploitation and exploration. Traditionally, these three stages are implemented using building blocks such as graph neural networks (GNN) as representations, variational inference (VI) or Gaussian process (GP) for uncertainty quantification, and analytical expressions as acquisition functions. To facilitate the integration of both domain-specific and general knowledge into various stages of this process, in this paper, we investigate which parts of this workflow can be augmented or replaced by large language models (LLM). To this end, we present **COLT** [1], a software library for **C**hemical **O**ptimization with **L**anguage- and **T**opology-based modules, and thoroughly benchmark the combination thereof. We found that *none* of the LLMs, no matter incorporated at what stage, can outperform the simple and fast Bayesian baseline with GNN and GP. As a remedy, we offer a new tuning recipe with direct preference optimization (DPO), where the optimization of synthetic properties can be used to increase the efficiency of the acquisition in real-world tasks.

The short answer to the question asked in the title is: *Not easily*.

## 1 INTRODUCTION: DRUG DISCOVERY AS SEQUENTIAL ACTIVE LEARNING

A drug discovery campaign—the endeavor to search, from the vast chemical universe, for a new chemical entity with some desired therapeutic efficacy—takes decades and billions of dollars and has its decision-making process traditionally reliant on human experts [1; 2]. When a human expert makes a decision to prioritize a certain compound(s) in the pre-clinical stage of a drug discovery project, their thought process, not without some oversimplification (regarding the multitask, constrained, and batched nature of the optimization), could be broken down as follows: first, an understanding of the current chemical space is constructed from existing medicinal chemistry data, as well as general knowledge distilled from years of training and experience (sometimes referred to as the *chemical intuition* [3]); second, this understanding is applied to the chemical space yet to be assayed, providing predictions with associated uncertainty; finally, by leveraging expectation and uncertainty, while balancing exploration vs. interpolation, she selects candidates to be synthesized and characterized, generating assay data to refine her belief. This process is carried out iteratively until a therapeutic candidate suitable for clinical trials is identified.

Bayesian optimization methods [4; 5] can work analogously to human experts when optimizing over the chemical space, with the aforementioned three stages corresponding to *representation*, *uncertainty quantification*, and *acquisition* in the active learning process. We can use graph neural networks (GNNs) [6; 7; 8; 9; 10; 11; 12] to represent the molecule (§ 2.1), variational inference (VI) [13] or Gaussian process (GP) regression [14; 15] to quantify the uncertainty of the predictions (§ 2.2), and analytical expressions as the acquisition functions (§ 2.3).

---

[1]code at: https://anonymous.4open.science/r/colt-6B75/

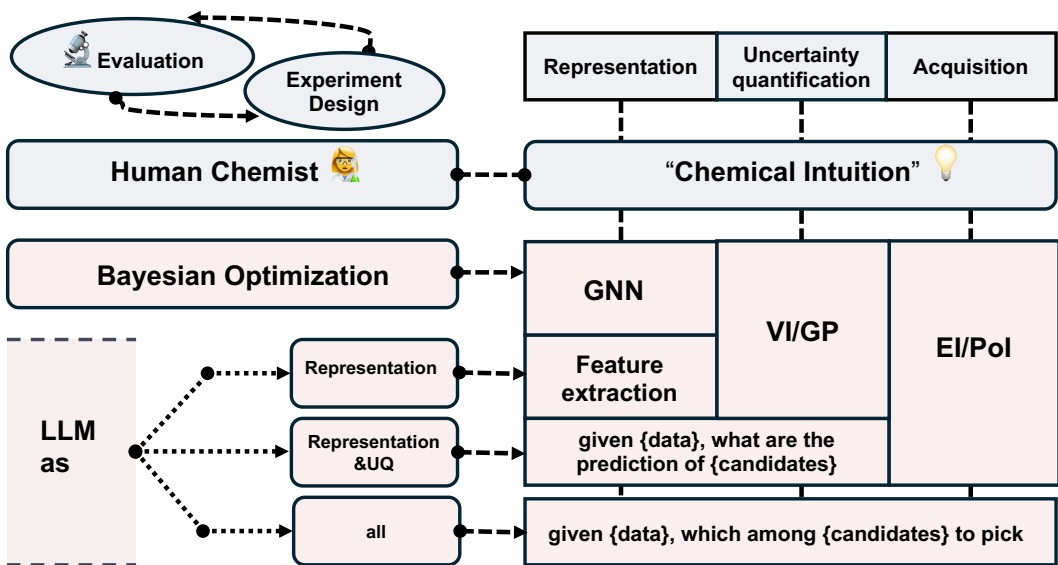

Figure 1: **Semantic illustration.** The experiment design in drug discovery can be broken down into the representation, uncertainty quantification, and acquisition stages, working analogously to the chemical intuition of human medicinal chemists. Bayesian optimization (§ 2) typically employs GNN as representation (§ 2.1), VI or GP (§ 2.2) as uncertainty quantification, and analytical functions (§ 2.3) for acquisition, which can be modularly replaced by LLMs using the example prompts.

Notably, the majority of early-stage drug discovery problems dwell in the *small-data* regime. Unlike the abundant cat pictures or blog posts readily scraped from the internet, in drug discovery, each data point is the result of costly and time-consuming wet lab experiments. Consequently, even the largest pharmaceutical companies rarely work with a candidate pool larger than 1 million compounds [16]—the number being smaller in magnitude for biotech startups. ChEMBL [17], the largest public chemical data repository, contains merely 15 million assays of various kinds over 1.8 million compounds. On one hand, this makes principled Bayesian models particularly well suited, thanks to their inherent regularization and uncertainty quantification. On the other hand, while working with graph-structured data provides useful inductive biases, incorporating these biases into such Bayesian, graph-based models remains challenging [18; 19], unlike LLMs, which can easily *absorb* language-structured knowledge by training on texts. In the chemistry domain, for instance, the success of LLMs has been demonstrated through numerous models fine-tuned for diverse modeling tasks [20; 21; 22; 23].

**Main contributions.** In this paper, we offer a comparison between the efficacy of *knowledge* versus *inductive bias* by evaluating the optimization efficiency of LLMs against that of BO with graph-based representation. First, we review the abstractions underlying the building blocks of Bayesian active learning and propose solutions to replace each of them with LLMs. To this end, we introduce a package for **C**hemical **O**ptimization with **L**anguage- and **T**opology-based models (COLT), which offers an easy-to-use API for molecular active learning. Using this program, we combinatorially benchmark the effect of LLMs when placed at various stages of active learning, and find that none of the solutions are as performant as the GNN + GP baseline. Offering a beacon of hope amidst the negative results, we provide a general tuning recipe based on *direct preference optimization* (DPO) [24], where learning occurs directly on the per-step *acquisition trajectory*, and find that tuning on synthetic datasets has a positive impact on optimizing real-world tasks.

**Limitations.** In this paper, we only consider the optimization on a finite candidate space, whereas the combinatorial chemical space is infinite and can be treated continuously [25]. Additionally, to enable rapid benchmarking with a limited computational footprint, we restrict our analysis to small datasets, resulting in high variance across the benchmark results. Similarly, we restrict our experiments to smaller, open-source LLMs.

## 2 BAYESIAN OPTIMIZATION (BO) FOR MOLECULAR ACTIVE LEARNING

First, we formalize the drug discovery campaign introduced in § 1. Given a finite pool of candidate chemical compounds, represented by chemical graphs $\mathcal{F} = \{\mathcal{G}_i, i = 1, 2, \dots N\}$, each associated with a *potency* $y(\mathcal{G}_i) \in \mathbb{R}$, of which a noisy observation is expensive to acquire, we are interested in finding the compound with the highest potency $\mathcal{G}^* = \text{argmax}_i\, y(\mathcal{G}_i)$ with fewest function evaluations. This can be achieved via *active learning*, starting with an empty portfolio $\mathcal{P} = \emptyset$, and in each round, choosing a compound from the candidate pool, $\mathcal{F}$, based on a decision informed by the current portfolio, $\hat{\mathcal{G}} = D(\mathcal{F}|\mathcal{P}), \hat{\mathcal{G}} \in \mathcal{F}$. Subsequently, we subtract this compound from the candidate pool to add it to the portfolio, $\mathcal{F} \leftarrow \mathcal{F} \setminus \{\hat{\mathcal{G}}\}, \mathcal{P} \leftarrow \mathcal{P} \cup \{\hat{\mathcal{G}}\}$.

Bayesian optimization (BO) is a powerful approach in active learning where the decision $D(\mathcal{F}|\mathcal{P})$ is constructed in a modular way. Firstly, we extract fixed $D$-dimensional features from the compounds to form a *representation*: $H = h(\mathcal{G}) \in \mathbb{R}^D$; secondly, this representation is used to construct a *predictive posterior distribution* with *uncertainty quantification*: $p(y|\mathcal{G}; \Theta), \Theta = \theta(H)$, whose parameters $\Theta$ is is mapped from the graph representations. Lastly, an *acquisition* function, $\alpha$, is applied on this predictive posterior to form a score, on the basis of which a decision is made: $\hat{G} = D(\mathcal{F}|\mathcal{P}) = \text{argmax}_{\mathcal{G} \in \mathcal{F}}\, \alpha(p(y|\mathcal{G}))$.

In sum, the acquisition trajectory represents a sequential decision process, where the probability of a given trajectory can be written as:

$$p(\mathcal{P} = \{\mathcal{G}_i\}) = p(\mathcal{G}_0) \prod_{t=0}^{t=|\mathcal{F}|} p(\mathcal{G}_t|\{\mathcal{G}_{t=0,\cdots,t}\}) \tag{1}$$

$$= p(\mathcal{G}_0) \prod_{t=0}^{t=|\mathcal{F}|} \int \mathrm{d}\,\Theta p(\Theta|\{\mathcal{G}_{t=0,\cdots,t}\})) P[\alpha(p(y|\mathcal{G}_t, \Theta)) \geq \alpha(p(y|\mathcal{G}_i, \Theta)), \forall \mathcal{G}_i \in \mathcal{F}]., \tag{2}$$

where the first equality (1) stands generally for active learning strategies surveyed in this paper (§ 2 and § 3), and the second equality stands only for traditional Bayesian optimization (§ 2).

### 2.1 REPRESENTATION: GRAPH NEURAL NETWORKS (GNN)

Graph neural networks (GNNs) have emerged to be the modern workhorse for graph representation. A GNN can be most generally defined as one adopting a layer-wise updating scheme that aggregates representations from a node's neighborhood $\mathcal{N}(v)$ (based on the edges $\hat{A}_{uv}$) and updates its embedding:

$$\mathbf{X}'_v = \phi(\mathbf{X}_v, \rho(\mathbf{X}_u, \hat{A}_{uv}, u \in \mathcal{N}(v))), \tag{3}$$

where $\phi, \rho$ are the *update* and *aggregate* function, repsectively. Omitting the nonlinear transformation step $\phi$, common to all neural network models, and assuming a convolutional *aggregate* function, $\rho = \text{SUM}$ or $\rho = \text{MEAN}$, a GNN layer is characterized by the aggregation/convolution operation that pools representations from neighboring nodes. This forms an intermediary representation $\mathbf{X}'$, which on a global level, with activation function $\sigma$ and weights $W$, can be written as: $\mathbf{X}' = \sigma(\hat{A}\mathbf{X}W)$. The primary difference among architectures lies in the choice of effective adjacency matrix, $\hat{A}$. The most classical examples include: graph convolutional networks [6] (GCN), which normalize $\hat{A}$ by the node in-degree, $D_{ii} = \sum_j A_{ij}$, and graph attention networks [11] (GAT), which take $\hat{A}$ to be the attention score;

$$\hat{A}_{\text{GCN}} = D^{-\frac{1}{2}} A D^{-\frac{1}{2}}; \hat{A}_{\text{GAT},ij} = \text{Softmax}(\sigma(\text{NN}(\mathbf{X}_i||\mathbf{X}_j))); \tag{4}$$

Stacking GNN layers, we can get a representation of the graph

$$H = \mathbf{X}^{(l)} = \underbrace{\sigma(\hat{A}\sigma(\hat{A}\sigma(\hat{A}...\sigma(\hat{A}}_{l \text{ times}} \mathbf{X} \underbrace{W...W)W)W)W}_{l \text{ times}}. \tag{5}$$

### 2.2 UNCERTAINTY QUANTIFICATION: VARIATIONAL INFERENCE (VI) AND GAUSSIAN PROCESSES (GP) REGRESSION

**Bayesian neural networks.** Under the Bayesian formalism, given sets of (graph, measurement) pairs as training data $\mathcal{D} = \{\mathcal{G}^{(i)}, y^{(i)}, i = 1, 2, 3, ..., n\}$, the probability distribution of the unknown

quantity of the measurement $y^{(n+1)}$ could be modelled with respect to the posterior distribution of the model $\mathcal{M}$ with parameters $\theta$ as:

$$p(y^*|\mathcal{M}, \mathcal{D}, \mathcal{G}^*) = \int p(y^*|\mathcal{G}^*, \Theta)P(\Theta|\mathcal{D}) \, \mathrm{d}\theta. \tag{6}$$

This integral, of course, is not tractable, and fully Bayesian methods using a Markov chain Monte Carlo-based sampling scheme can turn out to be prohibitively expensive with realistically sized datasets and models. We review two common techniques to approximate Equation 6.

**Variational inference on the parameter space.** Variational inference [13] turns the sampling problem into an optimization problem by assuming the distribution belongs to a specific family and and then optimizing its parameters to best approximate the true distribution. Concretely, on a parameter space [26], we assume such class of distribution to be a multivariate Gaussian with diagonal covariance matrix $q(\Theta) = \mathcal{N}(\boldsymbol{\mu}, \boldsymbol{\sigma})$, whose parameters can be optimized by minimizing the Kullback-Leibler (KL) divergence between the variational and true Bayesian posterior (also known as variational free energy):

$$\boldsymbol{\mu}^*, \boldsymbol{\sigma}^* = \arg\min_{\boldsymbol{\mu}, \boldsymbol{\sigma}} \mathcal{D}_{\mathrm{KL}}[q(\boldsymbol{\theta}|\boldsymbol{\mu}, \boldsymbol{\sigma})||P(\boldsymbol{\theta}|\mathcal{D})] = \arg\min_{\boldsymbol{\mu}, \boldsymbol{\sigma}} \mathcal{D}_{\mathrm{KL}}[q(\boldsymbol{\theta}|\boldsymbol{\mu}, \boldsymbol{\sigma})||P(\boldsymbol{\theta})] - \mathbb{E}_{\boldsymbol{\theta} \sim q(\boldsymbol{\mu}, \boldsymbol{\sigma})}[\log P(\mathcal{D}|\boldsymbol{\theta})], \tag{7}$$

**Gaussian process (GP) regression with deep kernel learning (DKL)** Within the deep kernel learning framework [27], a graph kernel can be defined by applying a standard kernel (such as the radial basis function, RBF) with parameters $\gamma$, to the output of GNNs.

$$k(\mathcal{G}, \mathcal{G}') = k_{\mathrm{RBF}}(\mathrm{GNN}(\mathcal{G}|\Theta), \mathrm{GNN}(\mathcal{G}'|\Theta), \gamma) \tag{8}$$

Equation 6 can then be written as:

$$p(\mathbf{y}^*|\mathcal{G}^*, \mathcal{D} = \{\mathcal{G}\}) \sim \mathcal{N}(\mathbb{E}[y^*], \mathrm{cov}(y^*)), \tag{9}$$

where:

$$\mathbb{E}[\mathbf{y}^*] = K(\mathcal{G}^*, \{\mathcal{G}\})[K(\{\mathcal{G}\}, \{\mathcal{G}\}) + \sigma_n^2 I]^{-1}\mathbf{y}; \tag{10}$$

$$\mathrm{cov}(\mathbf{y}^*) = K(\{\mathcal{G}^*\}, \{\mathcal{G}^*\}) - K(\{\mathcal{G}^*\}, \{\mathcal{G}\})[K(\{\mathcal{G}\}, \{\mathcal{G}\}) + \sigma_n^2 I]^{-1}K(\{\mathcal{G}\}, \{\mathcal{G}^*\}). \tag{11}$$

The neural network and the kernel parameters $\{\Theta, \gamma\}$ can be jointly optimized to produce a maximum likelihood fit to the dataset.

## 2.3 ACQUISITION FUNCTIONS

Once the model is trained, we can use the *predictive posterior*, $p(y^*|\mathcal{G}^*, \boldsymbol{\theta})$, of a given compound to prioritize compounds from a candidate pool by defining an *acquisition function* $\alpha$ and greedily selecting the candidate with the largest $\alpha$ for subsequent assaying. Popular choices include [28]:

*Probability of Improvement (PoI)*,

$$\alpha_{\mathrm{PI}}(\mathcal{G}^*|\mathcal{D}, \Theta) = 1 - \Phi_{P(y^*|\mathcal{G}^*, \boldsymbol{\theta})}(\max(y)), \tag{12}$$

characterizes the probability of the best current value, where $\Phi$ denotes the CDF of the corresponding distribution, and $\max(y)$ is obtained within the training set $\mathcal{D} = \{\mathcal{G}_i, y_i\}$

*Expected improvement (EI)*,

$$\alpha_{\mathrm{EI}}(\mathcal{G}^*|\mathcal{D}, \Theta) = \mathbb{E}_{P(y^*|\mathcal{G}^*, \boldsymbol{\theta})} \min\{\{y^*\} - \max(y), 0\}, \tag{13}$$

measures the expectation of improvement over the current best.

For tractable distributions such as the Gaussian family, these expressions can be evaluated analytically, though in general they can also be estimated via samples drawn from these distributions.

## 3 MODULARLY REPLACING BO COMPONENTS WITH LLMS

Large language models (LLMs)—neural networks composed primarily of transformer [29]—have shown surprising promise [30; 31; 32; 33] in representing and generating text-structured data and gained popularity very quickly in recent years. Their impressive generative ability led the field to describe them with personifying terms, such as *understanding*, and to employ them in seemingly impossible tasks, from reasoning [34] to planning [35].

In recent years, LLMs [31; 30] have been routinely incorporated into active learning pipelines, from hyperparameter search [36] to material design [37; 38]. Little attention, however, has been paid to dissecting which *stage* of the active learning process can LLMs be most effective in. To answer this question, and to compare the efficacy of LLMs with time-tested workflows, we propose three solutions to incorporate LLMs into molecular active learning by modularly replacing components outlined in § 2 by LLMs.

### 3.1 LLM AS REPRESENTATION

```
h = transformers.pipeline("feature-extraction")(molecule.smiles)
```

Replacing the GNNs (§ 2.1), one can use an LLM as a feature extractor to come up with the *representation* of molecules. The input of the LLM is a string representation of the graph, such as the SMILES string [39] common in molecular representation. This is consistent with the method Kristiadi et al. [38] used for material discovery.

Even before the era of LLMs, to represent molecules as strings and feeding them into transformer- or recurrent neural networks (RNN)-based models have long been used in various pipelines of molecular modeling, from property prediction to active learning [25], where pretraining on large ensembles of data has been proven as an effective avenue towards better performance [40]. We are interested in testing whether the added complexity and *knowledge* in LLM would help refine this representation.

### 3.2 LLMS AS REPRESENTATION AND UNCERTAINTY QUANTIFICATION

```
data = [molecule.smiles, str(molecule.y) for molecule in data]
prompt = f"""
Given a list of molecules with associated properties: {data},
what is the property of the molecule {new_molecule}?
"""
posterior_samples = [vllm.LLM().generate(prompt) for _ in range(N)]
```

The generation process of LLMs is intrinsically stochastic, and one can harvest the stochasticity of that process as a proxy of the uncertainty. Specifically, the predictions of a property of a molecule can be predicted by LLMs in a few-shot, in-context [41] manner, where a few examples are provided in the prompts, based on which a prediction of the desired property is made. This replaces both the *representation* (§ 2.1) and the *uncertainty quantification* (§ 2.2) stages of the Bayesian optimization. Ramos et al. [37] employed this method for quantifying the uncertainty of property predictions of materials. A natural challenge present here, as in all in-context learning methods, is the curse of the token limit, which allows very few samples to be included. When moving from the small-data to big-data regime, this can be solved via selecting in-context learning examples via similarity measures.

### 3.3 LLMS AS REPRESENTATION, UNCERTAINTY QUANTIFICATION, AND ACQUISITION

```
data = [molecule.smiles, str(molecule.y) for molecule in data]
candidates = [candidate.smiles for candidate in candidates]
prompt = f"""
In an active learning setting,
given a list of molecules with associated properties: {data},
which among the {candidates} to assay next to maximize the property
within the fewest rounds of assay?
"""
```

Finally, we can replace *all* building blocks introduced in § 2 with LLMs and let them make final choices of compounds to assay in the next round. When the pool of candidates is large, the length of the prompt will grow quickly. To enable token-efficient search among large candidate pools, we introduce a *tournament* search model, where we separate the candidates into groups, and iteratively compare the best candidate within groups. As such, if the token limit can originally fit only $N$ candidates, the tournament search model can allow up to $N^M$ candidates to be compared within $NM$ prompts.

# 4    COLT: A LIBRARY FOR MOLECULAR ACTIVE LEARNING.

```python
from colt import *
for representation in [GCN, LLMRepresentation]:
    for uncertainty in [VI, GP, LLMUncertainty]:
        for acquisition in [EI, LLMAcquision]:
            model = acquisition(uncertainty(representation()))
            trial = Trial(model=model, data=ESOL(), steps=100)
```

To rapidly benchmark the three strategies illustrated in Section 3, and to provide a platform for medicinal chemists to seamlessly integrate both traditional and language-based active learning approaches into their workflows – thereby accelerating the design of life-saving therapeutics – we introduce COLT, a software package for chemical optimization using language- and topology-based methods. The above is a illustration of the COLT library interface used to generate the results in § 5, where *representation* (§ 2.1 or § 3.1), *uncertainty quantification* (§ 2.2 or § 3.2), and *acquisition* function (§ 2.3 or § 3.3) are modularly abstracted.

**Speed.**    Designed for practical simulation, the efficiency in terms of wall time has been a focus of the design from the beginning. As such, in this PyTorch [42]-based package, the GNNs are implemented with generalized sparse matrix-matrix multiplication in deep graph library (DGL) [43], variational inference parallelized on GPU using Pyro [44] for variational inference, Gaussian processes regression with kernel interpolation [45] implemented in GPytorch [14], and fast deployment of LLMs with Huggingface [46] and vLLM [47].

**Similar packages.**    GAUCHE [48] performs Gaussian process regression using graph- and string-based kernels. Lapeft-Bayesopt [38] uses the representation from LLMs to perform Bayesian optimization to discover materials. COLT differs from these efforts in its ability to perform both traditional BO and LLM-based optimization, with both VI- and GP-based uncertainty quantification. It also stands out for its modular design and user-friendly interface.

# 5    EXPERIMENTS: A BENCHMARK OF COMBINING BUILDING BLOCKS.

Having introduced the machinery to modularly carry out the benchmark experiments on the molecular space, we benchmark the Bayesian graph- and language-based active learning strategies.

**Data.**    While we are interested in benchmarking the ability of our active learning algorithms to optimize the *potency* of compounds, high-quality, consistent potency data in the public domain is scarce due to the close-source nature of most drug discovery campaigns. Nevertheless, from a methodological point of view, the potency function $f$ is no more than a mapping from the graph structure to $\mathbb{R}$, and has the same *functional signature* as, and is dependent upon, the physiochemical properties of molecules. As such, we used the physical property datasets, ESOL [49], FreeSolv [50], and Lipophilicity from MoleculeNet [51]. The targets are normalized to be distributed within the range of $[0, 1]$, so the reported metrics are problem-agnostic [52]. Following the conceptual framework outlined in § 2, for all experiments, we start with an empty set, randomly select the first candidate, and iteratively refine the model based on the data already evaluated.

**LLMs are not as performant as more traditional models.**    Despite the high variance in the data, we notice that the best models uniformly arise from the composition of GNN, GP, and EI (except for the last row for the DPO model to be introduced). The more LLMs are involved in the active

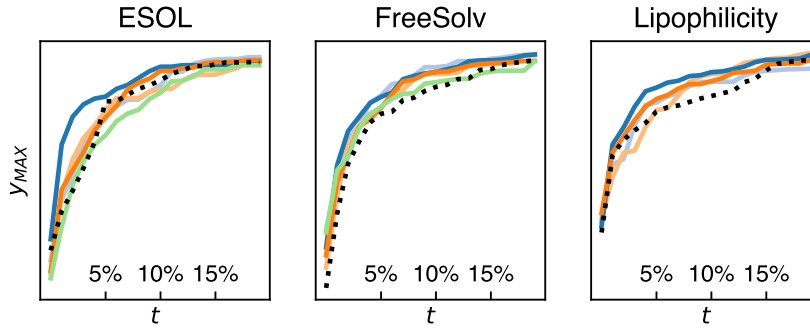

| Marker | Model | | | Normalized Cumulative Regret | | |
|---|---|---|---|---|---|---|
| | Rep. | U.Q. | Acq. | ESOL | FreeSolv | Lipophilicity |
| $\cdots$ | Random | | | $2.27 \pm 1.31$ | $2.38 \pm 1.57$ | $2.19 \pm 1.70$ |
| — | GCN [6] | VI | EI | $2.18 \pm 1.15$ | $1.62 \pm 1.16$ | $2.02 \pm 1.40$ |
| | | | PoI | $2.82 \pm 1.58$ | $2.18 \pm 1.15$ | $1.80 \pm 0.78$ |
| — | | GP | EI | $\mathbf{1.61 \pm 0.67}$ | $1.67 \pm 1.06$ | $1.54 \pm 1.07$ |
| | | | PoI | $2.39 \pm 0.38$ | $2.52 \pm 0.50$ | $1.10 \pm 0.43$ |
| — | GAT [11] | VI | EI | $1.29 \pm 0.77$ | $2.04 \pm 1.29$ | $2.65 \pm 2.00$ |
| | | | PoI | $2.41 \pm 1.16$ | $1.24 \pm 0.52$ | $1.60 \pm 1.22$ |
| — | | GP | EI | $2.08 \pm 1.03$ | $\mathbf{1.08 \pm 0.46}$ | $1.73 \pm 0.93$ |
| | | | PoI | $2.17 \pm 1.02$ | $2.11 \pm 0.57$ | $\mathbf{1.07 \pm 0.63}$ |
| — | Llama-8B [30] | VI | EI | $2.32 \pm 1.69$ | $1.96 \pm 1.36$ | $1.91 \pm 1.36$ |
| | | | PoI | $2.88 \pm 1.30$ | $2.06 \pm 1.40$ | $1.84 \pm 1.37$ |
| — | | GP | EI | $2.13 \pm 1.33$ | $1.92 \pm 1.19$ | $1.81 \pm 1.54$ |
| | | | PoI | $2.25 \pm 1.04$ | $2.01 \pm 1.09$ | $1.09 \pm 0.61$ |
| | bert [53] | GP | EI | $2.10 \pm 1.23$ | $2.03 \pm 1.30$ | $1.90 \pm 0.80$ |
| | Llama-8B | | EI | $2.40 \pm 1.30$ | $2.60 \pm 1.45$ | $1.95 \pm 0.43$ |
| | galactica-6.7b | | | $1.98 \pm 0.85$ | $1.70 \pm 0.48$ | $1.30 \pm 0.85$ |
| — | Llama-8B | | | $2.77 \pm 1.70$ | $3.00 \pm 1.24$ | $2.17 \pm 1.37$ |
| | Llama-8B$_{\text{description}}$ | | | $2.30 \pm 1.82$ | $1.95 \pm 1.48$ | $1.96 \pm 1.18$ |
| | galactica-6.7b [54] | | | $1.76 \pm 1.07$ | $2.04 \pm 1.54$ | $1.43 \pm 1.23$ |
| | galactica-6.7b$_{\text{description}}$ | | | $2.15 \pm 1.28$ | $1.87 \pm 1.44$ | $1.89 \pm 1.24$ |
| | ChemLLM-7B [20] | | | $3.41 \pm 1.16$ | $2.66 \pm 1.42$ | $2.51 \pm 1.48$ |
| | ChemLLM-7B$_{\text{description}}$ | | | $2.42 \pm 1.39$ | $2.86 \pm 1.75$ | $2.63 \pm 1.86$ |
| | DPO (§ 6) | | | $1.64 \pm 1.24$ | $\mathbf{0.91 \pm 0.38}$ | $\mathbf{1.03 \pm 0.45}$ |

Table 1: **Benchmark experiment on real-world dataset: The effect of replacing BO components with LLMs.** Normalized cumulative regret ($\downarrow$) with various *representation*, *uncertainty quantification*, and *acquisition* modules. Representative configurations are also plotted: the maximum value $y_{\text{MAX}}$ (averaged over 50 runs) plotted against the steps of acquisition. The random baseline is plotted in a dotted dark line. Confidence intervals are omitted in the figures for clarity.

learning pipeline, the worse the acquisition efficiency is—they can pass as feature extractor [38] when coupled with a GP (this is consistent with the findings of Ramos et al. [37]), achieving similar performance as a simple transformer [53]; when used as uncertainty quantification in an in-context manner, their performance deteriorates; and when used in an end-to-end manner to pick candidates directly, they act no different than the random acquisition function. At the same time, each graph-based, Bayesian model completes trials within a minute, whereas LLM-based models require by magnitude more time.

**The effect of domain-specific models.** There have been a plethora of LLMs for chemistry-related tasks [55], the most popular ones include Yu et al. [22]; Zhang et al. [20], which have been fine-tuned on instruction datasets [56]. Since they have "seen" more chemistry-related texts, it is reasonable to expect that they will perform better than the general-purpose models. This is the case for galactica [54], an LLM dedicated to science-related tasks, which outperforms the base Llama model.

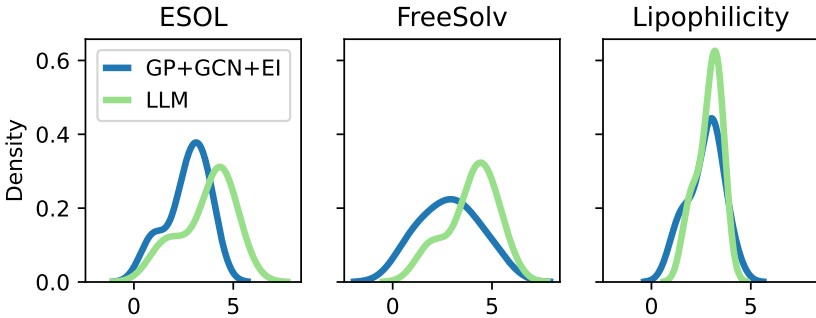

Figure 2: **Comparison of active learning algorithms on synthetic datasets.** Kernel density estimate (KDE) over the normalized cumulative regret ($\downarrow$) with GCN + GP + EI and LLM (`Llama-8B`) acquisitions over a range of synthetic targets defined in Equation 14. The candidate space is taken from the real-world dataset.

**Does knowing the target help?**    Again, the datasets employed in this section are reliably measured physiochemical properties. We decorate the prompt with the physical meaning of the targets in the trials marked with "description": aqueous solubility in ESOL, hydration free energies in FreeSolv, and the octanol/water distribution coefficient in Lipophilicity. Improvements are observed with descriptions added to the prompt.

**Have LLMs already seen this data?**    A risk ubiquitous in benchmarking LLMs on real-world data in the public domain is that there is no guarantee that the data was not in the training set. For instance, the datasets benchmarked here are explicitly included in the training set of Yu et al. [22]. Despite this risk, Table 1 illustrates that even with possible leakage, LLMs is still less competitive compared to GNN + GP initialized afresh. For a fairer comparison nonetheless, we synthesize an artificial target:

$$f(\mathcal{G}) = \sum_i \lambda_i K_i(\mathcal{G}, g_i), \tag{14}$$

where $K$ is a set of pre-defined graph kernels comparing the graph with certain fragments $g_i$, termed Morgan fingerprints [57], and $\lambda_i$ is taken from a Gaussian distribution. With each set of $\{\lambda_i\}$, we can correspondingly define a target. As shown in Figure 2, with synthetic dataset, the difference between the traditional, graph- and GP-based acquisition and the LLM is even more obvious.

**Experimental details and fixed degrees of freedom.**    Since the space of hyperparameter is vast and to find the best-performing model for each class is not in the scope of the paper, we prescribe a set of fixed experiment protocols for the graph-based Bayesian models benchmarked in Table 1. Within each round of acquisition, the neural network model is trained for 50 epochs with Adam [58] optimizer with learning rate $1e-3$ and L2 regularization factor $1e-5$; 8 samples are taken for all training and inference steps; and 1 attention head is used for GAT [11]. In this paper, we only consider exact GPs, leaving its variational counterpart [59] for future study. Even under these arbitrarily defined hyperparameters, these traditional models still perform better than more advanced LLMs.

## 6    DIRECT PREFERENCE OPTIMIZATION (DPO) PRETRAINING FOR MORE EFFICIENT OPTIMIZATION: TO THINK LIKE A GP, AND BETTER.

LLMs are routinely instruction-tuned [56] on supervised learning tasks to offer better predictions and "understandings" of certain candidate spaces. Without uncertainty calibration and carefully chosen acquisition functions, reliable predictions cannot be readily translated into efficient experimental design. In this section, we provide a general recipe to readily optimize the acquisition efficiency of LLMs in an end-to-end manner, i.e. taking over all of the representation, uncertainty quantification, and acquisition steps. We achieve this by learning from *all* of the choices (Equation 1) from successful trials driven by GNN + GP active learning algorithms.

Recall the sequential nature of the active learning—by assuming the greediness of Equation 1, we optimize the efficiency of the entire trial by optimizing that of each acquisition. We use direct preference optimization (DPO) [24] to achieve this goal. Specifically, note that in Table 1, the combination of GNN + GP is a uniformly strong baseline, we use this method to generate a cohort of $N$ trials, and find the most efficient trial, judged by the lowest normalized cumulative regret. Based upon our hypothesis, each step in the trial is efficient and worth encouraging. We therefore pair each acquisition step with a step generated from the random acquisition function as *preferred* and *dispreferred* samples, arriving at the policy objective:

$$\mathcal{L}_{\text{DPO}} = -\mathbb{E}(\mathcal{G} \in \mathcal{P}, \mathcal{G}_t^{\mathcal{GP}} = \text{argmax}_{\mathcal{G} \in \mathcal{F}}\, \alpha_{\text{EI}}(p(y_i)|\mathcal{G}), P(\mathcal{G}^{\text{random}}) = \text{Categorical}(\mathcal{G}_{>t}))$$

$$[\log \sigma(\beta \log \frac{\pi_\theta(\mathcal{G}_t^{\mathcal{GP}}|\mathcal{G}_{<t})}{\pi_0(\mathcal{G}_t^{\mathcal{GP}}|\mathcal{G}_{<t})} - \beta \log \frac{\pi_\theta(\mathcal{G}_t^{\text{random}}|\mathcal{G}_{<t})}{\pi_0(\mathcal{G}_t^{\text{random}}|\mathcal{G}_{<t})})] \quad (15)$$

Descending this objective leads to the LLM active learner not only to think *like* a GNN + GP algorithm, but also generating only the most successful trials. As shown in the last row of Table 1, this turns out to be a highly useful strategy. Starting from a particularly small model of `Qwen-0.5B` [60], we tune the acquisition efficiency to surpass that of the strong baseline of GNN + GP. Even though this particular experiment is small and only demonstrative, this only positive result amidst the discouraging performance of LLMs offers a promising avenue for tuning LLMs to directly make step-wise decisions in an active learning setting.

## 7 CONCLUSION.

In this paper, we abstract drug discovery as a sequential active learning process, and survey the Bayesian optimization (BO) building blocks that can lead to principled and efficient experiment design. Next, we review possible ways for these building blocks to be replaced by LLMs. A software package, COLT, is provided for carrying out active learning experiments and assessing its efficiency. Using this package, we thoroughly benchmark the effects of replacing BO building blocks with LLMs, and found that such replacement causes only slowdown and performance drop. This suggests that a large quantity of noisy knowledge (represented by LLMs) is not as effective as a principled model with appropriate inductive biases (BO + GNN). Nevertheless, under the framework of direct preference optimization (DPO), we provide a recipe for tuning LLMs directly for generating step-wise optimization decisions.

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
