# OpenReview forum: "Molecular Active Learning: How can LLMs Help?"
_ICLR.cc/2025/Conference — ICLR 2025 Conference Withdrawn Submission_

### Official Review · Reviewer_jRYu · 2024-10-20

**Soundness:** 2
**Presentation:** 2
**Contribution:** 1
**Rating:** 3
**Confidence:** 3

**Summary:**

The authors investigate how LLMs can be used in Bayesian optimization for molecular properties. They decompose the optimization loop into three parts: representation, uncertainty quantification, and acquisition. Using this framework, they iteratively replaced each of these parts with an LLM and compared its performance, using their modular framework COLT. The authors found that LLM-augmented campaigns could not outperform the state-of-the-art baseline models, except when direct preference optimization was introduced.

**Strengths:**

1) The authors have chosen to tackle a significant problem in the chemical sciences. Molecular discovery is a difficult problem and requires either expensive experimental evaluations or physics-based simulations. LLMs have shown promise in accelerating the evaluation of molecular properties, and understanding where LLMs can deliver the most impact in the discovery process is certainly useful knowledge.

2) The authors have proposed an interesting framework where LLMs can impact the discovery process in a Bayesian loop. Decomposing the process in this manner is a clear way to define the problem.

**Weaknesses:**

1) The authors intended to show whether LLMs could impact the molecular discovery process -- however, Ramos et al. (2023) and Kristiadi et al. (2024) have extensively demonstrated this already, so I am not sure what the contributions of the authors were. As the authors correctly identified, Kristiadi et al. (2024) has shown how LLM-based molecular representations can be used in molecular discovery, while Ramos et al. (2023) has shown how to obtain uncertainty from an LLM.

2) Kristiadi et al. (2024) proposes that the methods employed by Ramos et al. (2023) are not Bayesian in nature -- a direct comparison between the Bayesian baseline and the “stochasticity” of LLM-output does not seem sound to me.

3) While COLT was proposed as a software library, the code is not well-documented and is not installable.

4) The package seems very similar in capabilities to the package provided by Kristiadi et al. (2024), in that BO and LLM-based optimizations with GP-based UQ are also already features. The only extensions seem to be a graph-based representation and the inclusion of variational inference as a UQ estimate, which to me are not extensive enough to note the two packages as different.

5) The manuscript is generally unclear in the following aspects:
    - Key details regarding how the baseline GCN and GAT models were set up or trained are missing.
    - Information regarding the number of prompts used to generate the UQ estimates were missing.
    - The section investigating dataset leakage from the LLMs and synthetic data generation is unclear in terms of the problem setup, and the results are under-evaluated.
    - Even though DPO performed the best, which is a key takeaway of this paper, the reason why Qwen was chosen over all other LLMs is unclear. Additionally, the explanation of how the DPO experimental setup was done, and how DPO generated these results are unclear.
    - In Table 1, merging all the LLM methods into one line is extremely unclear, and it’s not straightforward to tell which LLM is performing the best.
    - Additionally, all the lines in Table 1 strongly overlap and it is difficult to tell which method is performing better. It almost looks to be like the differences between each model are not significant.
6) A minor point, but the code chunk at the header of Section 4 claims that LLM-based representations can be combined with Bayesian methods of UQ or acquisition, which I think is misleading.
7) GAT + VI + EI performs the best in ESOL, and the table numbers are incorrectly bolded.

**Questions:**

1) It appears that the GCN and GAT models were trained on each of the full tasks -- after which the representation was used for Bayesian optimization. This sounds to me like a weird baseline because this is contingent on having access to the full dataset. Could you explain this design choice? Why not use Morgan fingerprints instead as the baseline whose representations are output agnostic?
2) How do the baseline methods from Ramos et al. (2023) and Kristiadi et al. (2024) directly compare on the same task? Please ensure that the same exact methods (architecture, LLM choice, etc) were respected for a meaningful comparison.
3) I have general concerns about the LLMs used in this paper, and how they were used. As such, could you perform the experiments on all the below-mentioned models across the Bayesian loops and DPO, and across more tasks to get a better sense of how the LLMs could impact the performance?
    - Llama and Galactica are known to perform poorly on scientific tasks (see https://arxiv.org/abs/2402.13414, https://openreview.net/pdf?id=hSmn7BQZ2v, and https://arxiv.org/abs/2305.18365). Fine-tuning Llama seems to help (https://www.arxiv.org/abs/2409.06080), but this was not explicitly performed in this paper.
    - It seems premature to conclude that LLMs dont work if one does not compare the results from GPT-4, which is considered state-of-the-art.
    - Some smaller language models like ChemT5 or BioT5 have also been shown to outperform GPT4 in certain cases.
    - Qwen was used in DPO, but not in any of the other benchmarks, so it’s unclear if the benefits came from Qwen or from DPO alone.
    - Only three LLMs were sampled across your experiments. I think showing that these results apply across a wider variety of LLMs would help one of the main claims of the paper.
    - These results were only demonstrated for three tasks -- I also think it’s premature to draw conclusions from such a small sample.
    - The apparent “dataset leakage” problem from Yu et al. (2024) should have been demonstrated in the experiments -- this would have justified quantitatively why the synthetic dataset generation was necessary.
4) Could you explain exactly how you described the tasks in the prompts for the “description” subscript models? Showing the exact prompt in the Appendices section of the paper will help.
5) How confident are you with the uncertainty quantification methods proposed by Ramos et al. (2023)? I have my reservations about directly using LLMs for regression tasks (i.e to obtain real-valued numbers). In Section 3.2, these tokens are fed directly into the prompt. An ablation study to show that these numbers are actually correctly harnessed by your model in predicting the properties of a new molecule would be important.
6) How many samples are fed into the UQ part in Section 3.2, and how are they selected? The authors described that there is an issue with token length, but it was not clear to me exactly what they did to mitigate that issue.
7) Can you expand exactly what the motivation behind creating the artificial target as such is necessary? While I appreciate the mathematical definition of how the target is obtained, how this compares to the other datasets you compared against is not clear.
8) I have the same concerns for the definition of the DPO and a chemical interpretation of this task beyond just “helping the LLM to think like a GP”.
9) The difference in the performance on the synthetic dataset between traditional vs LLM approaches is “more obvious” -- but what does this actually mean? What is the source of this difference, and could you evaluate it?

---

> ### Author Response · Authors · 2024-11-13
> **Plan: Software package with better usability & more LLM variants**
>
> Many thanks, Reviewer `jRYu`, for your detailed, thorough, and critical feedback. We see that our manuscript can be significantly overhauled and improved based on your suggestions.
>
> # Clarification: GNNs are trained from scratch
>
> > It appears that the GCN and GAT models were trained on each of the full tasks
>
> The GNN models do not have access to the tasks---they are only trained on the acquisition trajectory so far. In other words, at the acquisition step $N$, their training set size is $N$, with no access to the rest of the unseen data. We will make this clearer in the manuscript.
>
> > Can you expand exactly what the motivation behind creating the artificial target as such is necessary?
>
> Firstly, because of the data leakage issues mentioned in Yu et al. 2024, the artificial targets would enable a fairer comparison. Secondly, we demonstrate that DPO can be trained on the acquisition trajectories of these artificial targets and perform on real-world datasets.
>
> # Contribution: Thorough comparison with first-principle graph-based models
>
> > Ramos et al. (2023) and Kristiadi et al. (2024) have extensively demonstrated this already
>
> Our paper is significantly distinct from these two papers, discussed in the Related Works section, in the following contributions:
> - None of them compared LLM active learning efficiencies with a first-principle baseline such as a GNN+GP.
> - We provide a thorough comparison among ways to incorporate LLMs whereas they focus on *one* way to incorporate LLMs in Bayesian active learning, Kristiadi et al. using it as representations and Ramos et al. as only uncertainty quantification.
>
> # Plan: Software package with better usability & more LLM variants
>
> > While COLT was proposed as a software library, the code is not well-documented and is not installable.
>
> We will improve the usability of this library during the discussion period and provide documentation and installation instructions.
>
> > I have general concerns about the LLMs used in this paper, and how they were used.
>
> Many thanks for providing ideas on other LLMs. We will explore more variants according to your suggestions.

---

> > ### Comment · Reviewer_jRYu · 2024-11-14
> >
> > Thanks for the quick turnaround in providing some clarifications.
> >
> > ## GNN Training
> >
> > I share a similar concern with reviewer poaA that graph models trained on very low amounts of data would likely be heavily overfit, so it appears to me that the initial stages of BO would choose bad points (i.e ill-informed selection). It would be interesting to see how the performance of the GNN changes as the BO campaign continues.
> >
> > ## Artificial Target
> >
> > I understand your motivation for constructing the artificial target. What I was referring to was why you chose this specific target, i.e why you constructed the artificial target in this manner, and if or how this artificial target resembles a chemical task. It does seem out of the blue to me because any artificial function could be evaluated, so I am curious why you decided to choose this artificial target.
> >
> > ## Comparison to prior works
> >
> > While it is true that GP+GNN baselines have not been explicitly performed in those two papers, I think it is well established that GP+GNN and GP+Fingerprints are already strong baselines, the latter of which was actually performed in Kristiadi et al. (2024). With that in mind, Kristiadi et al. (2024) does explicitly compare how different LLMs fare in active learning settings, so whatever was performed here was only a very minor difference from Kristiadi et al. (2024).
> >
> > Kristiadi et al. (2024)'s lack of an LLM-based UQ was specifically addressed in their paper as being incompatible with Bayesian methods, so I am curious about your answer to Q5 raised in my original review.
> >
> > ## LLM Variants
> >
> > To be clear, if there is a priority on the constraint on your resources and time, consider running Qwen > ChemT5 > other chemistry-specific LLMs. I understand that there are significant costs with running GPT-4, and that it is not possible to extract their embeddings which is required for your paper. However, I will maintain that the conclusions that one would make based on a benchmark study that does not consider a widely accepted state-of-the-art method would not be a strong.

---

> > > ### Author Response · Authors · 2024-11-14
> > >
> > > Thank you, Reviewer `jRYu`, for your suggestions. They are highly valuable for improving our manuscripts.
> > >
> > > # GNN Training: Planned plot
> > > > It would be interesting to see how the performance of the GNN changes as the BO campaign continues
> > >
> > > I think that this will be an interesting side result to show. We plan to include a plot on the trend of GNN performance as a function of acquired training data. This will perhaps also be a function of the exploration-exploitation strategy.
> > >
> > > > It does seem out of the blue to me because any artificial function could be evaluated, so I am curious why you decided to choose this artificial target.
> > >
> > > # Artificial target
> > >
> > > The choice is inspired by Gómez-Bombarelli et al. (2018) [1], where they constructed a linear combination of a few chemoinformatics descriptors. Since these descriptions can be accurately predicted by fingerprints, which are of higher dimension, we believe that the combination of fingerprints will be a deterministic mapping from the chemical structure and a rich yet realistic artificial targets for the active learning. If you have suggestions for better artificial targets we would also be more than happy to include them. Reviewer `tX3V` had suggested docking scores, which we are considering, although it might be too noisy.
> > >
> > >
> > > [1] Automatic Chemical Design Using a Data-Driven Continuous Representation of Molecules
> > >
> > > Rafael Gómez-Bombarelli, Jennifer N. Wei, David Duvenaud, José Miguel Hernández-Lobato, Benjamín Sánchez-Lengeling, Dennis Sheberla, Jorge Aguilera-Iparraguirre, Timothy D. Hirzel, Ryan P. Adams, and Alán Aspuru-Guzik
> > > ACS Central Science 2018 4 (2), 268-276
> > > DOI: 10.1021/acscentsci.7b00572
> > >
> > > > I think it is well established that GP+GNN and GP+Fingerprints are already strong baselines, the latter of which was actually performed in Kristiadi et al. (2024).
> > >
> > > # Comparison to prior works
> > >
> > > > I think it is well established that GP+GNN and GP+Fingerprints are already strong baselines, the latter of which was actually performed in Kristiadi et al. (2024).
> > >
> > > In the new experiments that we are preparing for the rebuttal, we are seeing that GP + GNN is a much stronger baseline than GP + Fingerprints, and is fairly robust. So there is direct utility for the practitioners in this field to use COLT to perform GP + GNN, to which we plan to add more documentation and examples to make it more user-friendly.
> > >
> > > > so I am curious about your answer to Q5 raised in my original review
> > >
> > > > An ablation study to show that these numbers are actually correctly harnessed by your model in predicting the properties of a new molecule would be important.
> > >
> > > I think it would be interesting to plot the in-context learning performance as a function of the volume of acquired data, together with a similar plot for GNNs, to see if the UQ afforded by LLM stochasticity is indeed meaningful.

---

### Official Review · Reviewer_tX3V · 2024-10-22

**Soundness:** 2
**Presentation:** 2
**Contribution:** 2
**Rating:** 3
**Confidence:** 4

**Summary:**

The authors investigated the effectiveness of using large language model (LLM) in the molecular activate learning. Specifically, they  evaluate LLM as a replacement to different modules in the active learning pipeline including molecule featurizer, uncertainty quantification, and acquisition. A software package called Chemical Optimization with Language- and Topology-based modules (COLT) is developed to carry out experiments. The authors revealed that the utilization of LLM in active learning does not bring performance improvement or speedup. To address the underwhelming performance of LLM, the authors proposed to use discrete preference optimization (DPO) to improve LLM's performance in active learning.

**Strengths:**

The approach used by the author to examine which part of molecular active learning can be replaced by LLM is interesting. The software effort to incorporate LLM for molecular active learning is also valuable.

**Weaknesses:**

1. Literature review is not comprehensive. Active learning/bayesian optimization for compound virtual screening from both academic ([ref1](https://pubs.rsc.org/en/content/articlelanding/2021/sc/d0sc06805e)) and industry ([ref2](https://pubs.acs.org/doi/abs/10.1021/acs.jcim.3c01938), [ref3](https://pubs.acs.org/doi/abs/10.1021/acs.jctc.1c00810)) are not included.

2. There are some mis-claims in the paper. For instance in line 82-83, pharmaceutical companies are definitely trying to perform virtual screening on libraries much larger than million-scale. A simple example of such large libraries is the Enamine database.

3. Experiments deviate from actual use cases of active learning in drug-discovery industry. The major goal of using active learning is to screen for hit. Therefore, a docking score would be a better option than labels in MoleculeNet. Authors should at least run experiment on the million-scale Emine HTS dataset included in [ref1](https://pubs.rsc.org/en/content/articlelanding/2021/sc/d0sc06805e).

4. More acquisition functions should be used as baseline such as greedy, upper confidence boundary etc.

**Questions:**

1. I am curious about the format of output when LLM is used as a feature extractor. Are those features vector of specific sizes? Can authors elaborate more?

2. What are the y-axis label and unit of the figure above Table-1?

---

> ### Author Response · Authors · 2024-11-12
> **Plan: More comprehensive literature review, more acquisition functions**
>
> Thank you, Reviewer `tX3V`, for your detailed feedback. During the discussion period, we plan to significantly improve our manuscript based on your suggestions.
>
> # Question: Targets for active learning
>
> > The major goal of using active learning is to screen for hit.
>
> > Therefore, a docking score would be a better option than labels in MoleculeNet.
>
> We would appreciate if you could kindly provide more evidence on the statement that hit-identification, rather than hit-to-lead optimization, is the primary use case of active learning in a drug discovery setting. Moreover, docking score is known to be extremely noisy. So we would be grateful if you could provide more clarification on why would that be a better target than the experimental measurements documented in MoleculeNet.
> # Plan: More comprehensive literature review, more acquisition functions
> > Literature review is not comprehensive.
>
> Many thanks for recommending new active learning papers. We will make sure to incorporate these in our literature review.
>
> > Pharmaceutical companies are definitely trying to perform virtual screening on libraries much larger than million-scale. A simple example of such large libraries is the Enamine database.
>
> Although _virtual_ screening has been routinely done on such databases, to the best of our knowledge, the biggest libraries for wet lab-based high-throughput screening is on the scale of millions of compounds. We will clarify this in our revised paper.
>
> > More acquisition functions should be used as baseline
>
> We will include greedy and UCB as our acquisition functions and compare with existing results.
>
> # Clarification
>
> > I am curious about the format of output when LLM is used as a feature extractor.
>
> The intermediate representation of LLM is used and a fixed-dimensional vector is produced as output.
>
> > What are the y-axis label and unit of the figure above Table-1?
>
> It is the normalized (unitless) measurement, as documented in the legend of Table 1. We will make this clearer in our revision.

---

### Official Review · Reviewer_newY · 2024-11-02

**Soundness:** 3
**Presentation:** 3
**Contribution:** 2
**Rating:** 5
**Confidence:** 4

**Summary:**

This paper addresses the challenge of active learning for molecular discovery, i.e. finding optimal molecules from large candidate libraries with minimum experimental evaluations. The authors formalize the active learning workflow into three modules: (1) representing molecules in vector form, (2) an uncertainty-aware regressor for predicting target properties, and (3) acquisition of the next candidate from the full library, balancing exploitation and exploration. The paper investigates whether some or all of these modules can be substituted with large language models to improve optimization efficiency. Empirically, the authors find that, even with domain-specific LLMs, no performance improvements can be observed. Instead, architectures with domain-specific inductive biases (i.e. GNNs for molecules) remain state-of-the-art for optimization.

**Strengths:**

* The study targets a highly topical problem with enormous implications in both academic and industrial settings.
* The paper, particularly the introduction and background sections, are very well written, making the paper clear and accessible.
* The LLM modules for molecular representations, molecular property prediction, and decision-making, are mostly well-designed. The design of their empirical evaluations is systematic and carefully crafted.
* The study effectively conveys an important message: LLMs may not always be suitable replacements for domain-specific models, particularly in those fields where knowledge and data are largely non-textual, and model architectures with strong inductive biases exist.

**Weaknesses:**

* The selected optimization tasks may not be adequate to rigorously test optimization algorithms. (a) The studied properties (solubility, lipophilicity) have relatively well-known structure–activity relationships, that expert chemists could predict, and that are potentially well-reflected in the training data of LLMs. This contrasts with real-life applications like drug discovery, where the structure–activity relationships are much harder to predict – for both expert chemists and ML models. (b) The candidate pools are small and biased.
* The optimization performance is measured by cumulative regret. However, this does not necessarily align with the primary goal of molecular discovery, which is to identify the optimal candidate(s). Alternative metrics that better capture this objective should be considered, and it would be interesting to analyze whether these metrics consistently correlate with cumulative regret.
* There is a lack of discussion on the statistical significance of findings and trends. Confidence intervals are omitted from the figures in Table 2. Given the large standard deviations in Table 1, it is unclear which of the discussed trends are actually significant. This should be addressed in the discussion. In my opinion, a generic sentence in the “Limitations” paragraph is not sufficient.
* The Direct Preference Optimization strategy lacks sufficient detail. How are the GNN–GP reference trials generated? Are they generated on the same optimization problem? If so, a direct comparison to other optimization strategies would be flawed. The need for initial GNN–GP-guided experiments to fine-tune the LLM, followed by a second campaign, would greatly increase experimental costs. In this scenario, all metrics should be compared over all performed experiments. Does the second, LLM-guided identify find any “good” candidate molecules that were not found in the initial GNN–GP trial? While I see that this strategy could be of value e.g. in a transfer learning setting or a multi-fidelity scenario, this paper does neither evaluate nor discuss these aspects.

**Questions:**

* The authors use Deep Kernel Learning as a domain model. Comparison with simpler domain-specific baselines, (e.g. fixed representations with a GP surrogates, see ref. 48), would place the findings in a broader context. What is the value of deep learning in the very-small-data regime? Do the inductive biases in GNNs make any difference?
* On p. 2, the discussion of the “small data regime” could be clearer. While the number of labeled data points is indeed very small (often only hundreds), the library of possible candidates can be extensive (up to billions, possibly more). Distinguishing these scales would clarify this section of the paper.
* The paper would benefit from investigating into the origins of the observed performance differences, e.g. through ablation studies or other systematic analyses. Are these differences due to poorly aligned representations, insufficient regression performance, lack of uncertainty calibration, or other factors?

---

> ### Author Response · Authors · 2024-11-13
> **Plan: Revised experiments & more experimental details.**
>
> Thank you, Reviewer `newY`, for your thoughtful and detailed review, and for highlighting the clarity and motivation of our manuscript. We are also grateful to your critiques, based upon which we plan to significantly improve our paper.
>
> # Plan: Revised experiments & more experimental details.
>
> > The selected optimization tasks may not be adequate to rigorously test optimization algorithms.
>
> > that are potentially well-reflected in the training data of LLMs
>
> We recognize that this is indeed a limitations of the datasets, hence we have synthesized artificial datasets to accompany the real-world datasets. In our revision, we plan to emphasize the synthesized datasets more as opposed to the smaller, real-world datasets.
>
> > The optimization performance is measured by cumulative regret. However, this does not necessarily align with the primary goal of molecular discovery, which is to identify the optimal candidate(s).
>
> We plan to also incorporate the rounds of acquisitions required to reach the best candidate as another metric. If you have other suggestions, we would make sure to incorporate those as well.
>
> > There is a lack of discussion on the statistical significance of findings and trends. Confidence intervals are omitted from the figures in Table 2.
>
> The experiments are inherently of high variance since they all start from randomly selected candidates. We will think of more rigorous ways to characterize the difference among trials.
>
> > The Direct Preference Optimization strategy lacks sufficient detail. How are the GNN–GP reference trials generated? Are they generated on the same optimization problem? If so, a direct comparison to other optimization strategies would be flawed. The need for initial GNN–GP-guided experiments to fine-tune the LLM, followed by a second campaign, would greatly increase experimental costs. In this scenario, all metrics should be compared over all performed experiments.
>
> We plan to provide more experimental details in the manuscript. Specifically, the reference GNN-GP trials are generated on synthetic datasets and tested on real-world datasets. We will make this clearer in the manuscript.
>
> > The authors use Deep Kernel Learning as a domain model. Comparison with simpler domain-specific baselines, (e.g. fixed representations with a GP surrogates, see ref. 48), would place the findings in a broader context
>
> We appreciate this suggestion. We will include simpler baselines in our revised experiments.

---

> > ### Comment · Reviewer_newY · 2024-11-18
> >
> > Thank you for the rapid response and the clarifications regarding your experimental plans for the revisions.
> >
> > > In our revision, we plan to emphasize the synthesized datasets more as opposed to the smaller, real-world datasets.
> >
> > Are synthesized datasets on "unphysical" properties (e.g.. the one defined in eq. 14) meaningful benchmarks? Can LLMs use their prior knowledge on physically meaningful structure–property relationships to obtain improved performance on these tasks? In my opinion, using large labeled libraries from virtual screening (e.g. molecular docking studies) would be a more realistic benchmark.
> >
> > > The experiments are inherently of high variance since they all start from randomly selected candidates. We will think of more rigorous ways to characterize the difference among trials.
> >
> > My key concern here is: If the optimization trajectories show such a strong dependence on the choice of initial conditions, how can we be sure that any of the discussed trends are statistically significant?

---

### Official Review · Reviewer_poaA · 2024-11-03

**Soundness:** 3
**Presentation:** 4
**Contribution:** 2
**Rating:** 6
**Confidence:** 4

**Summary:**

This paper presents what is effectively an ablation study between optimizing molecules with LLMs vs Bayesian optimization, finding that LLMs don't help very much, although fine-tuning does help.

**Strengths:**

- Scientific honesty: this paper asks an important question, finds that the less exciting "null hypothesis" is true, and still reports that. With almost every other paper at ICLR being a "use my method" paper, I found this refreshing
- The question asked is timely and important
- Experiments ask good questions (e.g. investigating domain-specific models and synthetic tasks after initially seeing poor results)
- Experimental methodology seems valid
- Paper is upfront about the limitations of its study
- Presentation of the paper is very nice

**Weaknesses:**

- Some important experimental details unclear (see "questions" section)
- The study does not address a question which is perhaps more relevant: how will LLMs perform _after some fine-tuning_. I think the paper should list this as a limitation, or potentially change the conclusion to "fine-tuning will probably be required to get strong performance"
- Some descriptions of BO seem slightly incorrect.
  - Learning a representation of a graph is an _optional_ step in BO, since one could use a model like a GP with a graph kernel that does not convert the graph into a vector. There are many such kernels supported in libraries like Grakel.
  - DKL is not an approximation to the posterior: it is a model whose exact posterior is analytically tractable.
- PoI acquisition function is not really used much because it is insensitive to the magnitude of the improvement. I recommend reading Chapter 7 of Garnett's 2023 BO textbook for details. I think that using the UCB acquisition function instead would be a more interesting choice.

**Questions:**

1. Experimental details:
  - How is the GCN trained, given that you start from no data?
  - How are GP hyperparameters set? Optimization performance is very sensitive to these parameters.
2. On line 370 you write: "require by magnitude more time". Is this a typo? Maybe you be "require an order of magnitude more time"
3. In table 1, what are the ± values? Standard error? Standard deviation? Something else?

---

> ### Author Response · Authors · 2024-11-12
> **Plan: UCB acquisition function, clearer experimental details**
>
> Dear Reviewer `poaA`,
>
> Thank you for your thoughtful and constructive feedback. We especially appreciate your encouragement for us to publish this interesting (primarily) negative results. We plan to significantly improve the clarity of our paper by incorporating your suggestions.
>
> # Clarification: Fine-tuning on acquisition trajectories
>
> > The study does not address a question which is perhaps more relevant: how will LLMs perform _after some fine-tuning_. I
>
> Our **Section 6, Direct Preference Optimization** provides a tuning recipe, albeit directly using the acquisition steps. In other words, we fine-tune the LLMs to _acquire_ like GPs. We argue that this provides a robust and performant recipe of fine-tuning, and traditional supervised fine-tuning has already been done in the `ChemLLM` and `galatica` models, which show no improved performance in acquisition efficiencies.
>
> # Plan: UCB acquisition function, clearer experimental details
>
> Following your suggestions, during the discussion period, we plan to repeat our experiments with UCB acquisition function. We also plan to clarify more experimental details:
>
> > - How is the GCN trained, given that you start from no data?
>
> The GCN is trained on all of the data point insofar acquired in the portfolio. Every time a new data point is added, the GCN training is continued.
>
> > - How are GP hyperparameters set? Optimization performance is very sensitive to these parameters.
>
> Following the deep kernel learning formulation, the GP hyperparameters are jointly optimized together with the GNN parameters.

---

> > ### Comment · Reviewer_poaA · 2024-11-13
> > **Reply to rebuttal plan**
> >
> > First, thanks for posting your rebuttal plan early: I've previously had negative experiences where authors perform a difficult set of experiments that don't really address the reviewer concerns, then get frustrated when reviewers don't change their scores. Posting early helps avoid such outcomes.
> >
> > Responding to your points:
> >
> > - Fine-tuning: I agree ChemLLM and galactica were fine-tuned in a supervised way, but I think this is different than _fine-tuning directly on your training data_. The tasks they fine-tuned on may not be very relevant to your task. I don't necessarily think you need to add this as a baseline, but it is the most natural baseline to add in my opinion. I would be open to increasing my score further if you added it.
> > - UCB acquisition function: add this if you want, but it doesn't seem critical to me (don't expect much of a score change from adding this)
> > - Training data: this might be a more important concern than you realized based on your short response. Since you start from no data and acquire points sequentially, it seems like you are training very large models (eg DKL or GCN) on tiny numbers of data points (<10). In this regime, there is probably massive overfitting and the results would depend heavily on regularization techniques like weight decay and early stopping. Is this something you have deliberately tuned in any way? It seems like something that should _hugely_ impact performance.

---

> > > ### Author Response · Authors · 2024-11-13
> > > **Fine-tuning in a supervised manner & comparing with pre-trained GNNs.**
> > >
> > > Many thanks for your response, Reviewer `poaA`.
> > >
> > > > [Fine-tuning on training data] is the most natural baseline to add in my opinion.
> > >
> > > After some thought, I believe that this is indeed a meaningful baseline to add. We will include a comparison with fine-tuning on training data.
> > >
> > > > In this regime, there is probably massive overfitting and the results would depend heavily on regularization techniques like weight decay and early stopping.
> > >
> > > We believe that GP is already an effective way to deal with small data and uncertainties, but the GNNs used here are uniformly small and regularized, which we plan to elucidate more in the revised experimental detail section. Inspired by your suggestion, we plan to also compare with a pre-trained GNN model.

---

> > > > ### Comment · Reviewer_poaA · 2024-11-14
> > > > **Doubt "GP is already an effective way to deal with small data and uncertainties"**
> > > >
> > > > Your previous response sounds good, but to be clear, your belief that "GP is already an effective way to deal with small data and uncertainties" is probably not true when the GP uses learned features (ie deep kernel learning, DKL). [Ober et al (2021)](https://proceedings.mlr.press/v161/ober21a.html) show pretty convincingly that DKL easily overfits to small datasets, and that GP marginal likelihood is _not_ sufficient to prevent this. If you are not familiar with this work I suggest skimming it, it really changed the way I think about DKL.

---

### Official Review · Reviewer_TxKs · 2024-11-04

**Soundness:** 3
**Presentation:** 3
**Contribution:** 2
**Rating:** 5
**Confidence:** 3

**Summary:**

This paper apply the active learning pipeline for drug discovery. By continuously selecting molecules from the pool to query their properties, high-quality molecules are obtained. Under this setup, three processes called GNN, VI and GP all have the potential to be replaced with LLMs and enhance the whole active learning performance. Authors explore the problems of whether, why and how? Finally, they propose a new tuning recipe that can be effective with LLMs for the drug discovery task.

**Strengths:**

1. The paper is well organized and written. Although the fields of active learning and AIDD do not naturally overlap, readers can quickly get their key points.

2. The proposed method is reasonable, and the three parts of active learning are indeed expected to be enhanced by introducing LLMs. In particular, the author provides code demonstrations. Intuitively, the method does not have significant loopholes.

3. The experimental results are comprehensive and the conclusions are reasonable. The proposed remedial solution seems to maximize the effectiveness of LLMs.

**Weaknesses:**

1. I am not very familiar with this specific task. Can the authors tell if this is a standardized and widely studied task? Can they provide some baseline methods for active learning-based drug discovery?

2. By reading this paper, I can understand the purpose of the task and the approach of the active learning part. And I think the author gives the conclusions related to LLMs, which is encouraged. However, my concern is that if active learning based drug discovery is not a widely researched topic, the significance or applicability of this paper will be diminished. For example, if researchers often use other means to complete drug discovery tasks and rarely use active learning (AL), then for this community, constructing an AL-based baseline is the primary goal. Because AL for drug discovery itself may have challenges worth exploring, which may be skipped and ignored. So, in short, I am not denying the novelty and experimental integrity of the research in this article. I just doubt whether this article will be of great value to this community.

3. It seems that ChemLLM is not better than LLaMA, which is counterintuitive for chemistry-related tasks. Can the authors explain why?

**Questions:**

Maybe I miss something, but why "1.61" in Table 1 is bolded not "1.29"? Can the authors provide more details?

---

> ### Author Response · Authors · 2024-11-12
> **Plan: More thorough introduction to motivate active learning in drug discovery**
>
> Dear Reviewer `TxKs`, many thanks for your detailed and constructive feedback, and for highlighting that our paper is well-organized and experimental results comprehensive.
>
> Please see here some clarification regarding your critiques and questions, which we will soon incorporate into the paper to improve clarity and readability.
>
> # Clarification: Significance of active learning (AL) in drug discovery
>
> >Can the authors tell if this is a standardized and widely studied task?
>
> > However, my concern is that if active learning based drug discovery is not a widely researched topic, the significance or applicability of this paper will be diminished.
>
> The overarching goal of the machine learning-assisted drug discovery is to improve the efficiency of decision-making, to find the most potent molecules within the fewest rounds of acquisitions, reducing the source requirements.
> Traditionally, in drug discovery, _supervised_ learning is used more ubiquitously, with the underlying assumption that one can use the predictions given by these models in a greedy manner to guide the search.
> In recent years, active learning approaches [Old references 37, 38; New references 1, 2] have been more frequently applied to molecular tasks. Leveraging uncertainty quantification to balance exploration and exploitation, active learning more directly optimizes the efficiency of acquisition and decision-making.
>
> One reason active learning is not being more popular in drug discovery is the lack of standard benchmarking testbeds and toolboxes, which is what COLT plans to address.
>
> > It seems that ChemLLM is not better than LLaMA, which is counterintuitive for chemistry-related tasks. Can the authors explain why?
>
> We believe that LLaMA is just a more capable foundation model that demonstrates more versatile utility in a wide range of tasks. We will design more thorough benchmarks to compare them.
>
> # Plan: More thorough introduction
>
> - In the discussion phase, we plan to include a more thorough introduction to motivate the design of COLT and the benchmark experiments.
>
>
> # References
>
> [1] Konstantin Gubaev, Evgeny V. Podryabinkin, Alexander V. Shapeev; Machine learning of molecular properties: Locality and active learning. __J. Chem. Phys.__ 28 June 2018; 148 (24): 241727. [https://doi.org/10.1063/1.5005095](https://doi.org/10.1063/1.5005095)
>
> [2] Chemical Space Exploration with Active Learning and Alchemical Free Energies. Yuriy Khalak, Gary Tresadern, David F. Hahn, Bert L. de Groot, and Vytautas Gapsys. Journal of Chemical Theory and Computation 2022 18 (10), 6259-6270 DOI: 10.1021/acs.jctc.2c00752

---

### Note · Authors · 2024-11-23

**Comment:**

Thank you so much again for your valuable feedback. And my sincerest apologies that we could not finish the planned experiment within the discussion phase, hence we are withdrawing this paper.

**Withdrawal Confirmation:**

I have read and agree with the venue's withdrawal policy on behalf of myself and my co-authors.